

# Unraveling genomic features and phylogenomics through the analysis of three Mexican endemic *Myotis* genomes

Edgar G. Gutiérrez[1], Jesus E. Maldonado[2], Gabriela Castellanos-Morales[3], Luis E. Eguiarte[4], Norberto Martínez-Méndez[1] and Jorge Ortega[1]

[1] Departamento de Zoología, Escuela Nacional de Ciencias Biológicas, Instituto Politécnico Nacional, Ciudad de México, Mexico
[2] Center for Conservation Genomics, Smithsonian's National Zoo and Conservation Biology Institute, Washington, D.C., United States of America
[3] Departamento de Conservación de la Biodiversidad, El Colegio de la Frontera Sur, Unidad Villahermosa (ECOSUR-Villahermosa), Villahermosa, Tabasco, Mexico
[4] Departamento de Ecología Evolutiva, Instituto de Ecología, Universidad Nacional Autónoma de México, Ciudad de México, Mexico

Corresponding author
Jorge Ortega, artibeus2@aol.com

## ABSTRACT

**Background**. Genomic resource development for non-model organisms is rapidly progressing, seeking to uncover molecular mechanisms and evolutionary adaptations enabling thriving in diverse environments. Limited genomic data for bat species hinder insights into their evolutionary processes, particularly within the diverse *Myotis* genus of the Vespertilionidae family. In Mexico, 15 *Myotis* species exist, with three—*M. vivesi*, *M. findleyi*, and *M. planiceps*—being endemic and of conservation concern.

**Methods**. We obtained samples of *Myotis vivesi*, *M. findleyi*, and *M. planiceps* for genomic analysis. Each of three genomic DNA was extracted, sequenced, and assembled. The scaffolding was carried out utilizing the *M. yumanensis* genome via a genome-referenced approach within the ntJoin program. GapCloser was employed to fill gaps. Repeat elements were characterized, and gene prediction was done via *ab initio* and homology methods with MAKER pipeline. Functional annotation involved InterproScan, BLASTp, and KEGG. Non-coding RNAs were annotated with INFERNAL, and tRNAscan-SE. Orthologous genes were clustered using Orthofinder, and a phylogenomic tree was reconstructed using IQ-TREE.

**Results**. We present genome assemblies of these endemic species using Illumina NovaSeq 6000, each exceeding 2.0 Gb, with over 90% representing single-copy genes according to BUSCO analyses. Transposable elements, including LINEs and SINEs, constitute over 30% of each genome. Helitrons, consistent with Vespertilionids, were identified. Values around 20,000 genes from each of the three assemblies were derived from gene annotation and their correlation with specific functions. Comparative analysis of orthologs among eight *Myotis* species revealed 20,820 groups, with 4,789 being single copy orthogroups. Non-coding RNA elements were annotated. Phylogenomic tree analysis supported evolutionary chiropterans' relationships. These resources contribute significantly to understanding gene evolution, diversification patterns, and aiding conservation efforts for these endangered bat species.

## INTRODUCTION

Whole genome sequences contribute to the identification of the genetic basis of specialization, of adaptation to ecological niches, and of different evolutionary events associated with natural history, origin, and persistence (*Zhao et al., 2009*; *Árnason et al., 2018*; *Coimbra et al., 2021*). Genome sequences are a fundamental to understand genome architecture, including the gene repertoire, the molecular mechanisms, and the adaptive evolution of the species (*Jones, Teeling & Rossiter, 2013*; *Ekblom & Wolf, 2014*; *Teeling et al., 2018*; *Armstrong et al., 2019*; *Jung et al., 2020*).

In the last decade, comparative genomic analyses have been used in different groups of mammals for the detection of genetic variation, gene rearrangements, evolution of gene families, functional genomics and phylogenomics, among others (*Gorbunova et al., 2014*; *Zoonomia Consortium, 2020*; *Chai et al., 2021*; *Yuan et al., 2021*). To date, there are about 50 bat genome assemblies available in the NCBI GenBank database, which have been used to identify unique bat evolutionary traits (*e.g.*, *Jebb et al., 2020*; *Nikaido et al., 2020*; *Moreno-Santillan et al., 2021*). Research focused on bat genomic comparisons has revealed signatures of adaptive evolution in genes related to metabolism, reproduction, visual function, longevity, the origin of flight under positive selection, expansion and contraction events of gene families associated with immune response, and chemosensory receptors (*Seim et al., 2013*; *Zhang et al., 2013*; *Hawkins et al., 2019*; *Gutiérrez-Guerrero et al., 2020*; *Yohe et al., 2021*). Other previous studies have been instrumental in unveiling signatures of parallel and convergent evolution in genes involved in the echolocation process (*Davies et al., 2012*; *Parker et al., 2013*; *Wang et al., 2020*).

Bats belong to the Order Chiroptera, representing one of the most diverse groups of mammals, with 1,474 recognized species (*Simmons & Cirranello, 2024*), and are ca. one fifth of all extant mammalian species (*Wilson & Mittermeier, 2019*). Bats are the sole group of mammals capable of performing active flight (*Jones & Teeling, 2006*), and most bat species possess a laryngeal echolocation system, allowing to effectively navigate, detect, locate, and capture their food (usually insects, but see below) at night (*Jones & Teeling, 2006*; *Dong et al., 2016*). These adaptations enabled bats to exploit a wide range of ecological niches, with the ability to feed on insects, crustaceans, frogs, small mammals, fish, nectar, fruits, and blood (*Kunz et al., 2011*; *Aizpurua & Alberdi, 2018*).

Interestingly, bats seem to harbor significantly more zoonotic viruses than other mammal groups but without manifesting any disease (*Calisher et al., 2006*; *Olival et al., 2017*), that seems to be related to a unique immune system and other unique adaptations that help counter infections (*Skirmuntt et al., 2020*; *Dutheil, Clinchamps & Bouillon-Minois, 2021*; *Ahn et al., 2023*). Some of these viruses of medical relevance for humans are Ebola, Nipah, Hendra, and the severe acute respiratory syndrome coronaviruses (*i.e.,* SARS-CoV) (*Li et al., 2005*; *Clayton, Wang & Marsh, 2013*).

Bats have genomes with a more limited size and constraint compared to other mammalian orders, exhibiting a genome around 2 Gb in size, comparable to that of various bird species (*Smith, Bickham & Gregory, 2013*; *Kapusta, Suh & Feschotte, 2017*; *Sotero-Caio, Baker & Volleth, 2017*). Bat karyotypes exhibit a wide range of diploid numbers, spanning from 2n = 14 to 2n = 62 in *Vampyressa* and rhinolophid species, respectively (*Kasai, O'Brien & Ferguson-Smith, 2013*). Notably, within the Vespertilionidae family, particularly in the genus *Myotis* possesses a karyotype with a diploid number of 44, which has remained unchanged and is therefore considered ancestral (*Bickham et al., 2004*; *Volleth & Heller, 2012*; *Sotero-Caio, Baker & Volleth, 2017*).

Vespertilionidae is the most species-rich family within Chiroptera, including more than 530 species (*Hill & Smith, 1984*; *Simmons & Cirranello, 2024*). Vespertilionid bats are found around the world, from tropical forests, semi-deserts, and deserts to temperate regions (*Stadelmann et al., 2004*; *Burgin et al., 2018*). In this family, the genus *Myotis* is very diverse, with ca. 139 in all the continents, except the polar regions (*Stadelmann et al., 2004*; *Simmons & Cirranello, 2024*). Most *Myotis* species are insectivorous, but some eat fish regularly or occasionally basis (*Siemers et al., 2001*; *Ma et al., 2003*; *Otalora-Ardila et al., 2013*; *Aizpurua et al., 2016*). *Myotis* bats have three main ecomorphotypes reflecting their feeding-foraging ecology (*Findley, 1972*; *Ruedi et al., 2013*; *Ghazali, Moratelli & Dzeverin, 2017*): "foliage gleaners", "trawlers" and "aerial netters" (*Findley, 1972*; *Morales et al., 2019*). These ecomorphotypes evolved several times in the different biogeographic regions where *Myotis* bats are found (*Ruedi & Mayer, 2001*; *Ghazali, Moratelli & Dzeverin, 2017*). Hence, the entire genus *Myotis* is a notable example of diversification, showcasing ecological and morphological convergences (*Stadelmann et al., 2004*; *Ruedi et al., 2013*; *Morales et al., 2019*).

A total of 139 *Myotis* species have been described, making it the most diverse genus of mammals (*Simmons & Cirranello, 2024*). In Mexico 15 species of *Myotis* are found (11% of all known *Myotis* species); three are considered endemic: *M. vivesi*, *M. findleyi*, and *M. planiceps* (*Ceballos, 2014*). The IUCN Red List of Threatened Species classifies *M. vivesi* as Vulnerable, and *M. findleyi* and *M. planiceps* as Endangered (*Arroyo-Cabrales & Ospina-Garces, 2016a*; *Arroyo-Cabrales & Ospina-Garces, 2016b*; *Arroyo-Cabrales & Ospina-Garces, 2016c*). These three species have restricted geographical distributions: *M. vivesi* is mainly found in islands of the coast of Gulf of California in the state of Sonora, and Baja California peninsula (*Avila-Flores & Medellín, 2014*); *M. findleyi* is only found in the Islas Marias in the Pacific Ocean, in front of Nayarit state (*Wilson, 2014*); *M. planiceps* is restricted to small and isolated montane areas in Nuevo León, Coahuila and Zacatecas states (*Arroyo-Cabrales et al., 2005*; *Jimenez-Guzman, 2014*), and was once considered extinct (*Baillie & Groombridge, 1996*). When grouping their feeding-foraging habits, *M. vivesi* falls under the trawlers, whereas *M. findleyi* and *M. planiceps* are catalogued as aerial netters.

Research on non-model organisms faces challenges due to limited genomic data availability. This issue is particularly pronounced in bats, where genomic resources are scarce for most species, hindering the study of their evolution. The lack of such resources has impeded research into molecular mechanisms and evolutionary processes in bats. To address this gap, we present the first genome sequence assembly for three

endangered endemic *Myotis* species. Leveraging the Illumina NovaSeq 6000 platform, we conducted high-throughput sequencing, assembled genomes, and identified transposable elements, revealing the presence of helitrons. Our study includes a comprehensive annotation of genes, prediction of non-coding RNA elements, and the generation of a cluster of orthologous genes by comparing with other *Myotis* genomes and a chiropterans phylogenomic analyses. This work contributes valuable insights into genome architecture, gene repertoires, environmental interactions, and evolutionary mechanisms in this diverse group of mammals.

## MATERIALS & METHODS

### Sample collection and sequencing

An individual of *Myotis vivesi* and *M. findleyi* specimens were collected in the field, while the *M. planiceps* sample was obtained from the collections at the Instituto Nacional de Antropología e Historia de México and provided by Dr. J. Arroyo-Cabrales. The individuals (*M. vivesi* and *M. findleyi*) underwent sedation using isoflurane. Subsequently, euthanasia was induced through inhalation within a nitrogen ($N_2$)-filled euthanasia chamber. Both procedures strictly adhered to the stipulations outlined in the official Mexican standard (NOM-033-SAG/ZOO-2014), which delineates the approved methodologies for euthanizing both domestic and wild animals. All samples were collected following the guidelines established by the American Society of Mammalogists for the ethical use of wild mammals in research (*Sikes, Thompson & Bryan, 2019*), and the procedures were carried out in accordance with Mexican Federal guidelines, license No. SGPA/DGVS/07992/20 issued by Secretaria de Medio Ambiente y Recursos Naturales (SEMARNAT) to Dr. Jorge Ortega. The samples obtained were preserved in liquid nitrogen, and subsequently stored at −80 °C, following *Yohe et al. (2019)* protocol for bat tissue collection for -omics analysis and primary cell culture. *M. vivesi* was collected in Coronado Island (26°08′11″N, 111°15′55″W), in the Gulf of California. *Myotis findleyi* was collected in María Cleofas Island (21°19′33″N, 106°14′12″W), in front of Nayarit state. *Myotis planiceps* was preserved in ethanol collected in Los Pinos, Arteaga (25°20′29″N, 100°47′24″W) Coahuila state; this site is a montane area on the edge of a small patch of piñon pine (*Pinus pseudostrobus*; *Arroyo-Cabrales et al., 2005*). The remaining tissue samples were deposited in the tissue bank "*Colección de tejidos de Vertebrados de la Escuela Nacional de Ciencias Biológicas*" (SEMARNAT No. DGVS-CC-328-CDMX/22).

Heart tissue samples from *M. vivesi* and *M. findleyi*, and a muscle tissue from *M. planiceps* were used to extract genomic DNA using the HMW-DNA extraction protocol for animal tissue with the PureLink and magnetic bead protocol v. 1.2 (*Kucka, 2020*). Agarose (1% concentration) gel electrophoresis was used to determine the gDNA integrity, and a Qubit Fluorometric Quantification (4 model; Thermo Fisher Scientific, Waltham, MA, USA) to determine the gDNA concentration.

Genomic DNA sequencing was performed using Illumina NovaSeq 6000 technology at the Novogene UC Davis Sequencing Center (Davis, CA, USA). Libraries were prepared by DNA fragmentation following the manufacturer's recommended protocol. Genomic

DNA was randomly cut into short fragments. The fragments obtained were end repaired by adding a single adenine base to form an overhang *via* an A-tailing reaction and further ligated with Illumina adapters (Table S1). Fragments with adapters were amplified by PCR, selected for size, and purified. The libraries were verified and quantified using a Qubit and real-time PCR, and a bioanalyzer was used for size distribution detection. Quantified libraries were pooled and sequenced according to effective library concentration and amount of data required. The libraries were selected based on the size of the fragments, allowing to discern between short (150 bp) and long (>150 bp) DNA fragments. Subsequently, the hairpin dimers and failed ligation products that formed during this process were eliminated.

## Genome assembly

The total number of obtained reads (in millions of sequences) from Illumina sequencing were determined through a FastQC analysis (*Andrews, 2010*). Reads were filtered to a minimum length of 150 bp and sequence coverage for all species was > 30X. The repair phase consists of detecting and correcting the errors in the reads. The procedure was carried out with Trimmomatic v. 0.3.9 (*Bolger, Lohse & Usadel, 2014*) considering the following steps: (1) detection of overlap between reads; (2) error correction through a consensus operation, and (3) elimination of sequencing adapters (from illumina sequencing) contained in reads. MaSuRCA v. 4.0.9 (*Zimin et al., 2013*) was run to perform *de novo* assembly of the filtered reads. The outcomes of these initial assemblies conducted *via* a de novo approach were evaluated using Quast v. 5.0.2 (*Mikheenko et al., 2018*; Table S2) for comparison.

After the construction of the contigs, scaffolding and gap-filling were carried out to connect contigs and obtaining longer sequences. First, we ran ntJoin v. 1.1.1 (*Coombe et al., 2020*), which is an assembly-driven scaffolder; this program needs a target assembly and at least one reference-assembly as input files. In this case, we used as a reference genome *M. yumanensis* (GCA_028538775.1). The genome sequence of *M. yumanensis* is one of the few Nearctic *Myotis* species available in the NCBI GenBank database. ntJoin uses minimizer graphs to produce a mapping between assemblies and reference information to generate scaffolds in the target assembly (*Coombe et al., 2020*). We subsequently ran GapCloser (*Luo et al., 2012*) for gap-filling using the filtered reads. Finally, the quality and integrity of the assembly was evaluated through the quantification of the genome assembly metrics (total length, total of contigs and scaffolds, L50, N50, among others) with the Quast software. Additionally, a visualization of each assembly contiguity and completeness was generated using assembly-stats v. 17.02 (*Challis, 2017*). The evaluation of the integrity of the assembly was performed by searching for single copy orthologous genes with BUSCO v. 5.4.3 (*Manni et al., 2021*) and the Mammalia_odb10 database that currently includes 9226 mammalian orthologous genes, available on OrthoDB (https://www.orthodb.org; *Kuznetsov et al., 2023*).

## Characterization of repetitive elements

Characterization of transposable elements (TEs) included in each assembly were predicted, annotated, and masked using RepeatModeler and RepeatMasker v. 4.1.3 (*Flynn et al., 2020*;

*Smit, Hubley & Green, 2022*). RepeatModeler identified families of transposable elements throughout the genome (*Flynn et al., 2020*). For RepeatMasker we used two databases, one database is a bat-specific custom transposable elements generated by manual curation (*Jebb et al., 2020*), the second contains mammalian transposable elements available on RepBase, as well as a manual curation of TE sequences from the Zoonomia project (*Zoonomia Consortium, 2020*; *Paulat et al., 2022*). After obtaining the masked genomes and files containing the repetitive elements found in our genomes, we calculated the substitution rate (through the Kimura substitution level; *Kimura, 1980*) of the TE's, using the *calcDivergenceFromAlign.pl* and *createRepeatLandscape.pl* scripts provided in the RepeatMasker package. The Kimura substitution model (*Kimura, 1980*) was then used to estimate the relative age and history of transposition of the TE's sequences in the three genomes.

## Gene prediction and functional annotation

Protein-encoding genes in the three *Myotis* genomes were comprehensively annotated using two approaches. The first approach is carried out through different types of protein/transcripts evidence (by homology) that is available in databases. The second is known as *ab initio* gene prediction. These two gene annotations approaches were performed in the Maker pipeline v. 2.31.10 (*Holt & Yandell, 2011*). The masked genomes (output files of RepeatMasker) and the available extrinsic evidence were used as input files to perform gene prediction by homology. We used transcriptomic and proteomic data from *M. lucifugus* (GCF_000147115.1) and *M. myotis* (GCA_014108235.1) available from the NCBI GenBank database as extrinsic evidence. In the second approach, the *ab initio* gene prediction was carried out using the output data from the first run (.*gff3* extension out file from Maker result) through the Augustus v. 3.4.0 (*Stanke & Morgenstern, 2005*) and SNAP v. 2006-07-28 (*Korf, 2004*) within the Maker pipeline. Additionally, the Annotation Edit Distance (AED) score was calculated to determine the accuracy of each gene annotation, with the *AED_cdf_generator.pl* script, also available within the Maker pipeline. The AED score is employed to assess the precision of a genome annotation. It is derived from merging annotation metrics related to accuracy and comprehensiveness. Annotations with AED scores at or below 0.50 are deemed satisfactory, while those with scores at or below 0.30 are indicative of exceptionally high-quality annotations (*Holt & Yandell, 2011*).

Functional annotation of the structural final gene sets was performed by detection of protein domains using three approaches. The first was carried out using InterproScan v. 5.62–94.0 (*Jones et al., 2014*) and each of the gene predictions were compared against gene annotations available in the InterPro database (*Blum et al., 2021*). The InterPro database integrates predictive information about protein function from many resources providing a description of the protein function (*Blum et al., 2021*). The second approach is based on protein homology through BLASTp v. 2.13.0+ (*Camacho et al., 2009*). This package requires a database to compare and detect the best match for each gene analyzed. The functional inference analysis was generated using the protein sequence databases UniProtKB/Swiss-Prot (*UniProt Consortium, 2023*) and NCBI RefSeq (*O'Leary et al., 2016*). The third approach corresponds to the search for putative biological functions of

the structural predicted genes were assigned by compared against the Kyoto Encyclopedia of Genes and Genomes (KEGG; *Kanehisa et al., 2023*). Low quality genes of less than 50 amino acids and/or exhibiting premature termination were removed.

## ncRNA annotation

To predict and annotate all non-coding RNA (ncRNA) sequences included in our genomes, we used INFERNAL v. 1.1.4 (*Nawrocki & Eddy, 2013*) and tRNAscan-SE v. 2.0.12 (*Chan et al., 2021*). Five main types of ncRNA were annotated in these analyses: transfer RNA (tRNA), ribosomal RNA (rRNA), micro-RNA (miRNA), small nucleolar RNA (snoRNA), and small nuclear RNA (snRNA). The tRNA genes were annotated by tRNAscan-SE, with parameters assigned for eukaryotic genomes, and the sequences of the snoRNA, snRNA, rRNA and miRNA genes were inferred with INFERNAL. We used the Rfam v.14.9 database (*Kalvari et al., 2020*) as a reference to perform the aforementioned analysis.

## Homology inference

Orthologous gene clustering was conducted employing two distinct approaches. The first approach involved consolidating our three recently acquired gene annotations with an additional five gene annotations linked to *Myotis* species: three species distributed in the Old World (*M. myotis*, GCF_014108235.1; *M. daubentonii*, GCF_963259705.1, and *M. davidii*, GCF_000327345.1/) and two species distributed in the New World (*M. brandtii*, GCA_000412655.1, and *M. lucifugus*, GCA_000147115.1). These annotations were sourced from the NCBI RefSeq database. For the second approach, we combined our three *Myotis* gene annotations, with 37 other bat genome annotations most of these provided by *Moreno-Santillan et al. (2021)*; as well as two gene annotations that used as outgroups (Table S3). These two strategies were instrumental in the grouping of orthologous genes. To perform orthologous gene clustering, we utilized Orthofinder v. 2.5.4 (*Emms & Kelly, 2019*) with default settings. In addition, we harnessed the capabilities of OrthoVenn3 (*Sun et al., 2023*) to explore and visualize the resulting ortholog clusters.

## Phylogenomic tree reconstruction

We sought to unveil the phylogenetic relationships among the recently acquired annotations of endemic Mexican *Myotis* species and the bats with available genomic resources by reconstructing a phylogenomic tree. In this endeavor, we focused on the clustering of single-copy orthologous genes. Briefly, the procedure entailed amino acid-level sequence multiple alignment, incorporating the sequences from 274 single-copy orthogroups. These orthogroups resulted from the grouping of 40 bat species, with two mammals serving as an outgroup. The MAFFT v. 7.520 software (*Katoh & Standley, 2013*) was employed to perform this multiple alignment, and subsequent corrections were made to rectify misaligned regions using Gblocks v. 0.91b (*Talavera & Castresana, 2007*). We conducted a search to identify the optimal evolutionary model that best fits our dataset using ModelTest-NG v. 0.1.7 (*Darriba et al., 2020*). Finally, the genomic tree was reconstructed through a Maximum Likelihood (ML) approach, using the IQ-TREE v. 2.2.5 software (*Minh et al., 2020*) with branch support estimated through 1,000 bootstrapping replicates. The branch lengths depicted in the tree signify their respective coalescent units.

## RESULTS

For each species more than 200 million of reads were obtained (Fig. S1). The raw reads from the three sequencings, are accessible to the public under BioProject PRJNA1021227 in the SRA database of the NCBI. FastQC analysis showed a total of 238,848,022 reads by each sense (5′–3′and 3′–5′) for *M. findleyi* (Fig. S1), followed by 232,122,197 reads for *M. planiceps* and 216,183,530 reads for *M. vivesi*, with the lowest number (Fig. S1). In general, the reads are 150 bp in length and have a Phred quality ≥ 30 (Fig. S1). Therefore, sequencing with Illumina Novaseq 6000 was efficient and accurate, placing all reads in the optimal range. The number of reads eliminated during the post-trimming process did not exceed 0.003% of the total reads in each file (Fig. S2). The discarded reads encompassed sequences that did not exhibit concordance between the forward and reverse strands of the double-stranded DNA, displayed low quality, or contained elements associated with the adapters utilized during the sequencing process. A second FastQC analysis was performed to compare between pre- and post- filtering (Fig. S3).

The final length of each genome assembly ranged from *M. findleyi* with a total of 2,050,536,070 bp (2.05 Gbp) in length in 60,544 scaffolds (Fig. 1; Fig. S4); followed by the *M. vivesi*, with a total length of 2,064,249,348 bp (2.06 Gbp) in 56,186 scaffolds (Fig. 1; Fig. S4) and *M. planiceps* with 2,080,496,215 bp in 256,303 scaffolds (Fig. 1; Fig. S4). The GC content was 42.50% in the *M. planiceps*, 42.71% in *M. vivesi,* and 42.91% in *M. findleyi* (Fig. 1; Fig. S5). The N50 for *M. vivesi* and *M. findleyi* were N50 = 91,830,945 bp and N50 = 92,497,855, respectively (Fig. 1), while for *M. planiceps* was lower, 83,203,680 bp. L50 statistic was seven in *M. vivesi* and *M. findleyi* and nine in the case of *M. planiceps* (Table S4). Through the detection of orthologous genes with BUSCO and the Mammalia_odb10 database, we detected 56.5, 91.5 and 93.8% of complete single-copy genes in *M. planiceps*, *M. findleyi*, and *M. vivesi*, respectively (Fig. 1). The specific number of complete orthologous genes detected were 5,219, 8,438, and 8,654 in *M. planiceps*, *M. findleyi*, and *M. vivesi*, respectively (Fig. 1). Contrastingly, the variability in the count of missing orthologous genes is evident across species. Specifically, for *M. vivesi*, this figure stands at 4.2%, while for *M. findleyi*, it slightly increases to 5.5%. Notably, in the instance of *M. planiceps*, this percentage rises to 32.5%, marking the highest proportion noted within the assemblies (Fig. 1).

In the *M. vivesi* genome, the analysis using RepeatMasker resulted in the masking of approximately 36.4% of the total sequence, which corresponds to TEs (Fig. 2), in *M. findleyi*, a total of 35.8% of the genome's total length was masked (Fig. 2) and in *M. planiceps* 28.7% of the complete genome sequence was masked (Fig. 2). Within the three genomes the most frequent TEs, are long interspersed nuclear elements (LINEs) representing 14.4%, 13.2%, 10.6% of the complete genome length in *M. vivesi*, *M. findleyi*, and *M. planiceps*, respectively (Fig. 2). The second most frequent type of sequences are short interspersed nuclear elements (SINEs), with a total of 937,607 (5.20%) sequences in *M. vivesi* genome, followed by 880,584 sequences (including 6.02%) in *M. findleyi* genome, and 5.9% of *M. planiceps* (Fig. 2). The less abundant repetitive elements included smaller percentages (<4%) of simple repeats, satellites, and low complexity elements (Fig. 2).
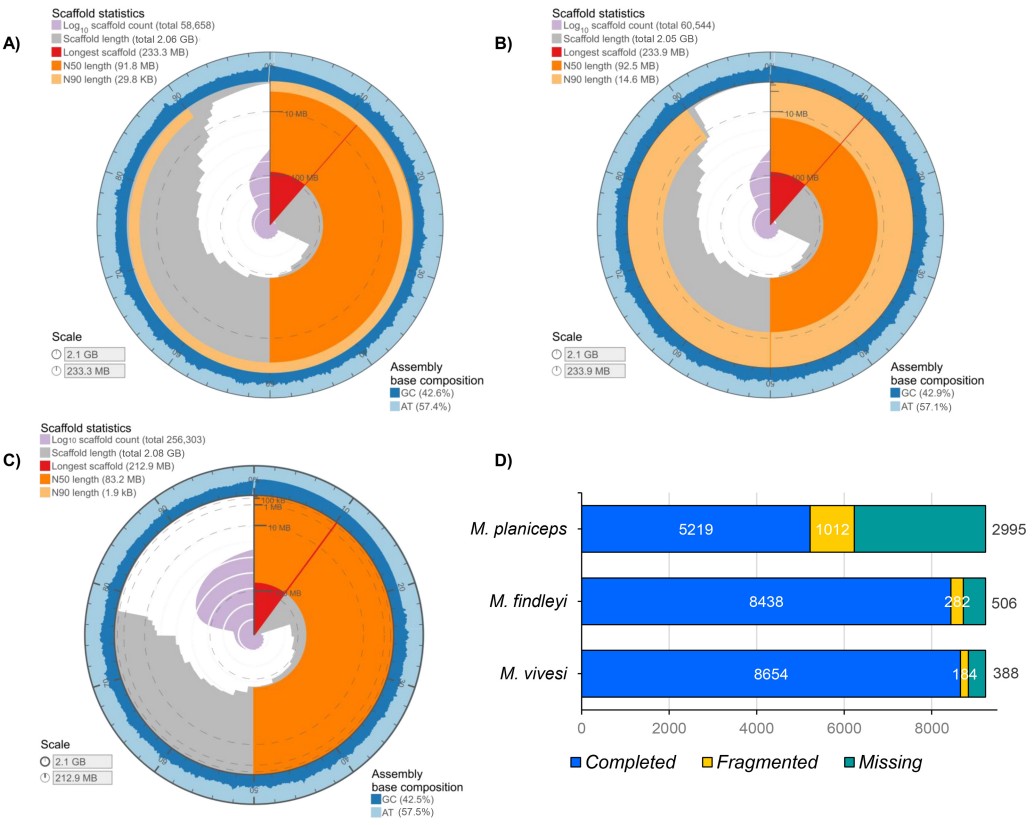

**Figure 1** Graphical representation of the genome assembly statistics of (A) *M. vivesi*, (B) *M. findleyi*, and (C) *M. planiceps*. The contiguity and genome assembly metrics of each of the three *Myotis* species are visualized as a circle, representing the total length of each assembly (2 Gb). The graphs were generated and visualized at: https://github.com/rjchallis/assembly-stats. (D) The integrity analysis of each genome assembly through the identification of orthologous genes with BUSCO and the Mammalia_odb10 database.

The Kimura distance-based copy divergence analyses showcased a similarity in divergence of the most prevalent TEs sequences compared to the consensus TE sequence across the three *Myotis* genomes (Fig. 2). Within the genome profiles of the three *Myotis*, a notable predominance of retrotransposons was observed in contrast to other elements (*e.g.*, DNA transposons). The Kimura distance analysis indicated a relatively recent transposition activity, aligning with the maximum peaks of the mainly TE families (Fig. 2). These peaks are related to bursts of TE transposition events within each genome. Noteworthy divergence peaks were particularly prominent in LINEs, Helitrons, and SINEs elements, while comparatively smaller divergence peaks appeared in DNA elements and LTR elements (Fig. 2). Remarkably low K values detected among the analyzed genomes pointed to the existence of recent, potentially active copies of elements. The profiles are characterized by a significant peak of relatively recent LINEs elements (including Helitrons and SINEs) and a more recent peak of DNA transposons (with the minimum *K* values). Among the oldest
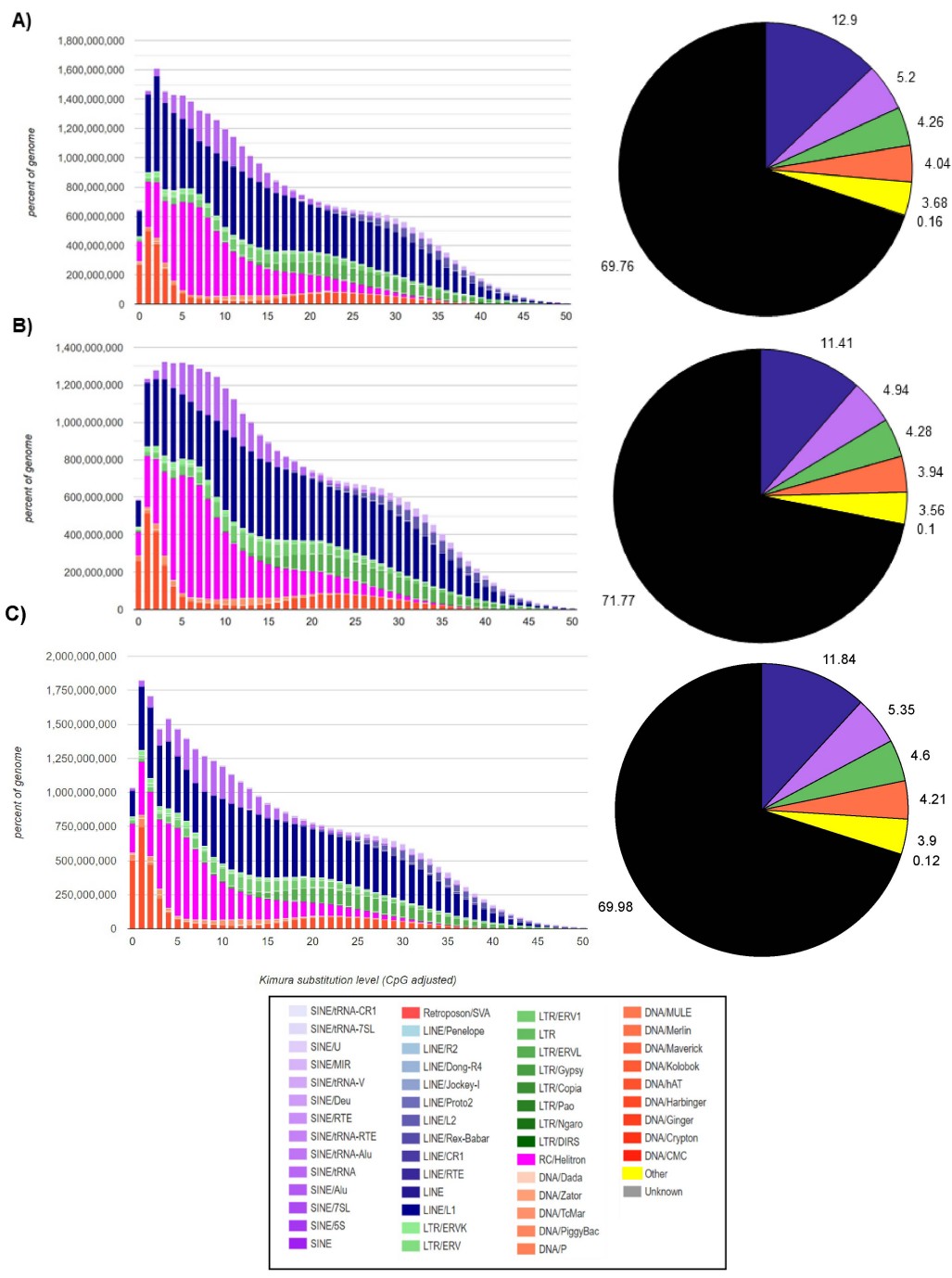

**Figure 2 Repeat landscape obtained by the Kimura substitution level related to the occupied proportion of the genome (left) and the fraction (right; pie chart) of the TE families (SINEs, LINEs, LTR retrotransposons, Helitrons and DNA elements) in the three genomes.** For each element, the graph shows the sequence divergence from its consensus with the Kimura distance (*x-axis*) in relation to the genome percent of each TE families (*y-axis*). Elements with older transposition activity are shown on the right side of the graph, while most recent transposition activity are depicted on the left side. The graphs correspond to the genomes of (A) *Myotis vivesi*, (B) *M. findleyi*, and (C) *M. planiceps*.

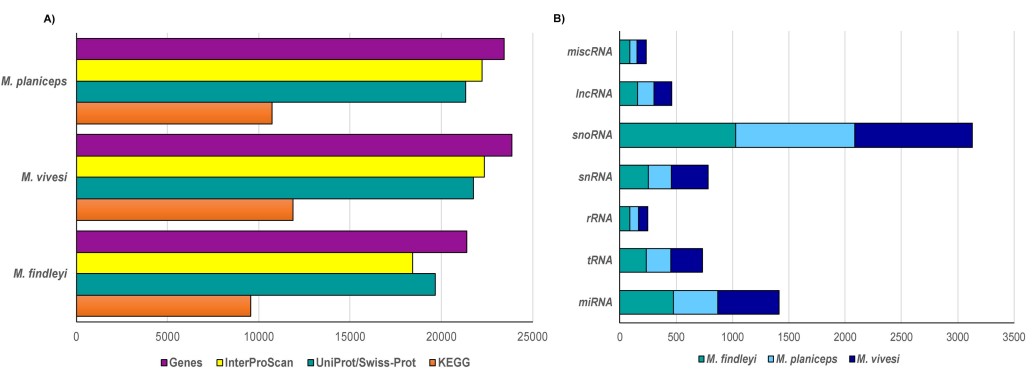

**Figure 3 Number of predicted genes in the three genomes of the Mexican endemic species of *Myotis*.** Structurally and functionally annotated using the indicated databases. (A) Protein coding genes, and (B) the main types of non-coding RNA genes.

(with the > *K* values) identifiable elements within the three *Myotis* genomes are LINEs, SINEs, and LTRs.

We performed a structural gene prediction detecting 21,394, 23,444, 23,851 genes in *M. findleyi*, *M. planiceps*, and *M. vivesi*, respectively (Fig. 3). Regarding the functional annotation using the InterProScan and UniProtKB/Swiss-Prot databases, these values changed, annotating between 86.2 and 91.9% of genes associated with a function in *M. findleyi*, followed by 91.2 and 93.7% in *M. vivesi* (Fig. 3; Table S5) and 91 and 94.8% in the *M. planiceps* (Fig. 3; Table S5). In addition, the KEGG annotation score 49.9, 45.7 and 44.7% positive hits in *M. vivesi*, *M. planiceps*, and *M. findelyi*, respectively (Fig. 3; Table S5; Fig. S6). Furthermore, the cumulative fraction of the AED distribution ranged from 0 to 1, where more than 90% of the annotations have an AED score less than 0.5, portraying the satisfactory annotation precision for each of the three *Myotis* genomes (Fig. S7).

Prediction of ncRNA elements resulted in the annotation that ranged from 2000 to 2500 ncRNA sequences, while the largest number of sequences was snoRNA in all three cases (1028—*M. findleyi*; 1041—*M. vivesi*; and 1059—*M. planiceps*, Fig. 3). The miRNA sequences exhibited varying counts among the species: *M. planiceps* displayed 397 sequences, *M. findleyi* revealed a count of 492, while *M. vivesi* showcased the highest count at 542 sequences. The counts of tRNA differed across the species: *M. planiceps* exhibited 218 sequences, *M. findleyi* showed 263, and *M. vivesi* presented the highest count at 284 sequences (Fig. 3). The snRNA elements counted for 205, 247, and 324 sequences in *M. planiceps*, *M. findleyi*, and *M. vivesi*, respectively. Among the ncRNA annotation, rRNA genes were found to have the lowest frequency, ranging from 10 in *M. planiceps*, 13 sequences in *M. vivesi*, and 19 elements in *M. findleyi* (Fig. 3).

We implemented orthologous gene clustering using eight species of the genus *Myotis* (three Mexican endemics and five other available species in the NCBI RefSeq database) which generated a total of 20,820 orthogroups. The total number of genes associated in orthogroups was 170,943, corresponding to 95.2% of the total number of genes analyzed, indicating that only 4.8% of these were unassigned genes. Of the total orthogroups, 9,398

contain at least one representative in each of the complete set of the eight species analyzed. Of the above set, 4,789 correspond to single-copy orthogroups (Table S6). The number of singletons has the lowest proportion with 324 (Table S6). Ortholog clustering analysis, focusing on the association between Old and New World species, has generated orthogroups containing at least one representative of each subset, shedding light on similarities between species. Specifically, when examining New World species, a substantial 9,648 orthogroups were identified, indicating a notable clustering effect within this subset. Similarly, when associating the three endemic Mexican *Myotis* species with those distributed in the Old World, another 9,657 orthogroups were shared, signifying important genetic similarity (Fig. 4). Nevertheless, pairwise comparison between the three endemic species under scrutiny and the *Myotis* species from the New and Old World starkly reveals disparities in shared orthogroup numbers (Fig. 4). A higher proportion of shared orthogroups is observed among New World species. In contrast, a lower ratio characterizes the relationship between the species in our study and those distributed in the Old World (Fig. 4). These findings are supported in heat map analysis, which vividly illustrates a strong relationship of orthologous genes between phylogenetically related species (Fig. S8).

By clustering orthologous genes from our focal species with genomic datasets derived from 39 other species including two species from the supraorder Euarchontoglires were employed as outgroups, which included humans and mice, we successfully identified a total of 247 orthogroups. Subsequently, a multiple sequence alignment was performed using a concatenated sequences for each species, yielding a comprehensive matrix comprising approximately 197,975 amino acids for every species included in our analysis. In this endeavor, we employed the JTT+I+G4 evolutionary model, selected as the most suitable model for our data based on the outcomes of ModelTest-NG. Our findings pertaining to the reconstruction of the phylogenomic tree using a ML analysis provide robust support for an early divergence event between the suborders Yinpterochiroptera and Yangochiroptera. This inference is underscored by a bootstrap support value of 100 (Fig. 5). The species of interest included to the Yangochiroptera lineage, within the Vespertilionidae family, and they form a well-supported cluster with other members of the *Myotis* genus, with robust branch support ranging from 94 to 100% bootstraps.

## DISCUSSION

We assembled, annotated, and described the first reference-based drafts genomes for three Mexican endemic *Myotis* species that will be useful for studies on adaptive evolution, molecular mechanisms and gene rearrangement, among others. The genomes ranged from 2.05 to 2.08 Gb, similar to genomes in other *Myotis* species (*i.e.*, 2.03 Gb in *M. lucifugus* (GCA_000147115.1), 2.06 Gb in *M. davidii* (GCA_000327345.1), 2.11 Gb in *M. brandtii* (GCA_000412655.1), two different genome lengths in *M. myotis* (2.0 and 2.29 Gb; GCA_014108235.1, GCA_004026985.1, respectively), and in *M. yumanensis* (1.95 and 2.05 Gb; GCA_028538775.1, GCA_028536395.1, respectively)). Species belonging to other vesper bat have similar genome lengths (*i.e.*, *Eptesicus fuscus* (2.01 and 2.03 Gb; GCA_027574615.1 and GCA_000308155.1, respectively), *Antrozous pallidus* (2.1 Gb; GCA_027595915.1), *Corynorhinus townsendii* (2.1 Gb; GCA_026230055.1)).
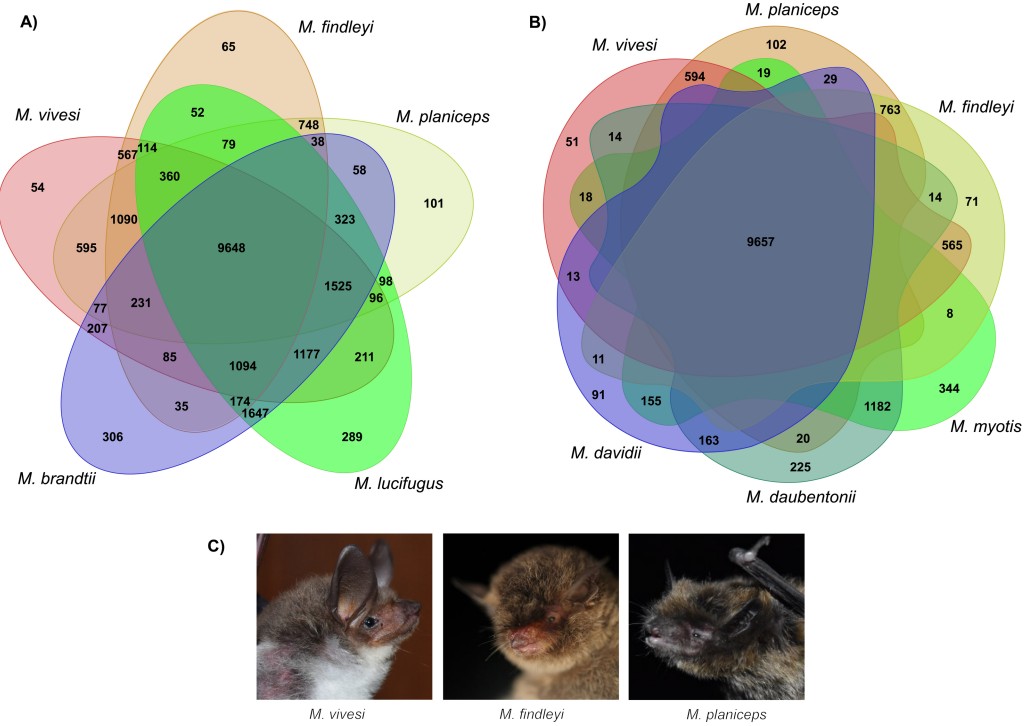

**Figure 4** **Clustering analysis of orthologous genes in eight *Myotis* species.** (A) Venn diagram illustrates the total orthogroups generated from the association of three *Myotis* species endemic of Mexico (*M. planiceps*, *M. vivesi* and *M. findleyi*) together with two other *Myotis* species from the New World (*M. brandtii* and *M. lucifugus*). (B) Orthogroups shared between the three Mexican endemic species and three *Myotis* species distributed in the Old World (*M. Myotis*, *M. davidii* and *M. daubentonii*) are represented. Gene annotations were obtained from NCBI RefSeq. The Venn Diagrams were created in https://orthovenn3. bioinfotoolkits.net/ and subsequently refined using Adobe Illustrator v. 27.0. (C) The Mexican endemic *Myotis* species analyzed in this study. Photos by: Edgar G. Gutiérrez and Mercedes Morelos.

Variation in length between genomes is common among species in a genus and, even in different individuals of the same species. Variation in genome length in mammals can be explained by different numbers in repeat sequences, including satellite DNA, TEs (*e.g.*, LINEs and SINEs) and ribosomal genes (*Lindblad-Toh et al., 2005*; *Biémont, 2008*; *Kapusta, Suh & Feschotte, 2017*). In addition, it has been reported that centromeres and the Y chromosome play an important role in the differences in genome size both within and between species (*Biémont, 2008*). Another likely explanation is artifacts resulting from the sequencing technology, the depth of sequencing, and the employed assembly algorithms. The future implementation of diverse levels of sampling and replicates and the adoption of long-read sequencing technologies such as PacBio and Nanopore will play a crucial role in elucidating the scope and underlying factors contributing to both within- and between-species variations (*Sims et al., 2014*).

Transposable elements constituted ~29–36% of the total length of each of our assemblies, encompassing SINEs, LINEs, LTR retrotransposons, DNA transposons and other low frequency elements including simple repeats and satellites. The proportions of TEs

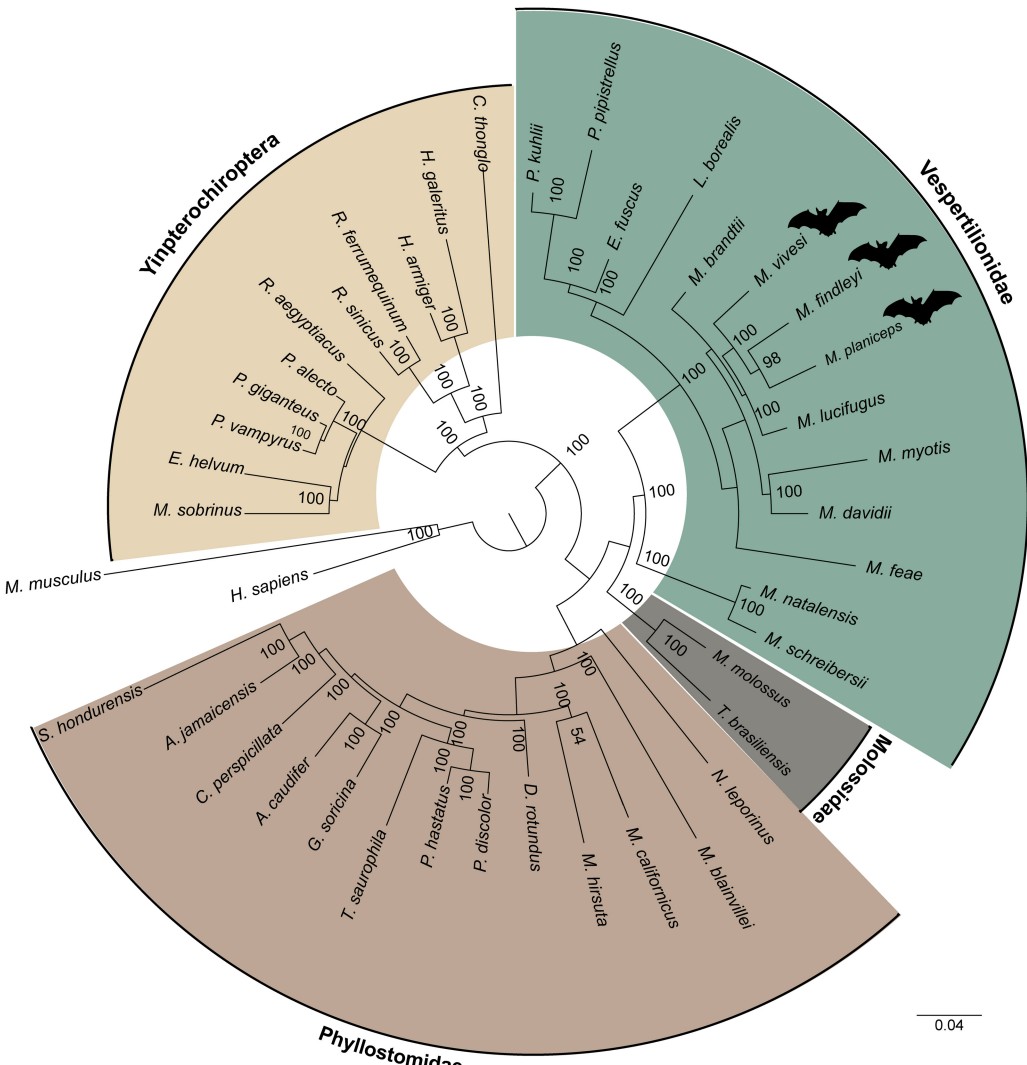

**Figure 5** **A maximum likelihood phylogenomic tree was reconstructed for a concatenated dataset comprising 42 species with 247 orthogroups.** The phylogenomic tree was reconstructed using the JTT+I+G4 model of sequence evolution. The numbers on the nodes represent bootstrap values. The colors distinguish the different families, indicated on the figure, except for the families within the suborder Yingpterochiroptera which we represented as single group.

families in these three *Myotis* species were lower compared to other Vespertilionid bats: *Eptesicus fuscus* (40.58%), *M. davidii* (42.61%), *M. lucifugus* (47.16%), *Pipistrellus pipistrellus* (47.95%), and others (*Osmanski et al., 2023*), but we should remember that the quality and depth of the assembly could affect the precision of the TE annotation (*Osmanski et al., 2023*). Across other bats, the distribution of TEs varies notably. For instance, in *Molossus molossus* (Molossidae), TEs constitute about 41.91% of the total genome length, while in *Carollia perspicillata* (Phyllostomidae), this proportion is approximately 31.10%. In the case of *Micronycteris hirsuta* (Phyllostomidae), TEs encompass a significant 46.71% of the genome. When examining bats belonging to the Yinpterochiroptera suborder, the TE
proportions tend to be approximately one-third of the genome size. For instance, *Pteropus vampyrus* (Pteropodidae) displays a TE proportion of around 30.37%, *Rhinolophus sinicus* (Rhinolophidae) exhibits 34.97%, and *Rousettus aegyptiacus* (Pteropodidae) shows a TE proportion of 35.79% (*Osmanski et al., 2023*). In other mammals TE proportions are even higher, reaching above 50% of the complete genome are common (*i.e.*, the bison, *Bison bison* (50.12%), the woodland dormouse, *Graphiurus murinus* (63.21%), the aardvark, *Orycteropus afer* (74.45%), the latter is an example of a mammal that has most of the genome covered by TEs discovered to date (*Osmanski et al., 2023*)). In addition, we found Helitrons in the three *Myotis* genomes. The Helitrons elements are represented by a history of transposition that began about 30-36 mya within the Vespertilionidae family (*Thomas et al., 2011*).

The Kimura distance-based comparative analysis, which estimates the divergence of TE sequences, and relates to the history of transposable activity, reveals that the three transposable accumulation profiles are shaped by recent bursts of TEs, such as DNA transposons, and retrotransposons, such as LINEs, and SINEs. These profiles represent waves of insertion and excision of TEs families to the genomes (*Platt II et al., 2014*). Our TE profiles results are in line with recent TEs activity reported in other vespertilionid bats (*e.g.*, *Lasiurus borealis*, *Eptesicus fuscus*), including other *Myotis* species (*e.g.*, *M. myotis*, *M. lucifugus* and *M. brandtii*), which have experienced several recent, independent bursts of DNA transposons and retrotransposons (*Platt II, Mangum & Ray, 2016*; *Jebb et al., 2020*; *Osmanski et al., 2023*; *Paulat et al., 2023*).

Nevertheless, accumulation profiles of TEs exhibit distinct variations among other vesper bats, such as *Miniopterus natalensis* and *Pteronotus parnellii*, but also in species of the Yinpterochiroptera suborder, *e.g.*, *Pteropus vampyrus*, *Rousettus leschenaultii*, and *Eidolon helvum*. These TE accumulation profiles unveil a contrasting trend wherein the ages of TE insertions reflect extended periods of genomic inactivity or very low activity. This phenomenon is particularly evident in the older waves of TE insertions, underscoring the intermittent nature of genomic dynamics in these species (*Platt II, Mangum & Ray, 2016*; *Nikaido et al., 2020*; *Osmanski et al., 2023*; *Paulat et al., 2023*). Although the origin of TEs is not exactly known, horizontal transfer could be a likely mechanism of TEs origin and spread between genomes (*Kofler et al., 2018*; *Paulat et al., 2023*).

Overall, between 2,000 and 2,500 non-coding RNA (ncRNA) sequences were annotated. Among these sequences, a larger number of snoRNA sequences were identified in the three species under examination, each with comparable counts: 1,028 in *M. findleyi*; 1,041 in *M. vivesi*; 1,059 in *M. planiceps* (Fig. 3). In contrast, previous studies have reported highly variable quantities of each ncRNA type in other bat species, including some *Myotis* species (*Platt II et al., 2014*; *Yuan et al., 2015*; *Jebb et al., 2020*; *Mostajo et al., 2020*). Our results are consistent with *M. myotis*, showing a higher abundance of snoRNA elements, followed by miRNAs, and a lower abundance of rRNA (*Jebb et al., 2020*). However, our results differ from *M. lucifugus*, *M. brandtii*, and *M. davidii*, which exhibited a higher abundance of lncRNA elements, followed by miRNAs, and a lower abundance of snoRNA elements (see *Mostajo et al., 2020*). Discrepancies also arise to the findings in *M. ricketii*, wherein *Yuan et al. (2015)* identified a notably higher count of rRNA elements in contrast to any other

category of non-coding RNA. There are few genomic or transcriptomic studies in bats that have annotated non-coding RNA elements. Consequently, our understanding regarding the vast collection of untranslated RNA and its fundamental role in various regulatory and cellular functions is limited, including their roles in differentiation and development (*Bartel, 2004*; *Mostajo et al., 2020*).

The prediction and annotation of orthologous genes is a fundamental requirement for many genomic analyses. This has enabled the prediction of gene function, the evolution of gene families, the reconstruction of phylogenetic trees, the identification of differences in the genes underlying the phenotypes that are likely to be a consequence of adaptations or may contribute to adaptations that evolved in organisms (*Kirilenko et al., 2023*). The number of structurally annotated genes and genes associated with a biological function varied between each of our genomes (*e.g.*, InterProScan annotation, from 18,439 to 22,351 in *M. findleyi,* and *M. vivesi*, respectively). This variation aligns with findings reported in genomic annotation inferences for other bat species (*Myotis myotis* (21,303 genes), *Molossus molossus* (20,221 genes), and *Phyllostomus discolor* (20,953 genes) (*Jebb et al., 2020*) and other mammalian species (*i.e.*, *Ursus maritimus* (17,189 genes), *Tursiops truncatus* (15,868 genes) and *Acinonyx jubatus* (16,969 genes); (*Hecker & Hiller, 2020*)). The composition of the data set as well as technical factors—including differences in the contiguity and quality of each set—influence on the accuracy of ortholog inference (*Trachana et al., 2011*; *Kirilenko et al., 2023*). Through the association with gene expression (*i.e.,* transcriptomic data), the quality and precision of the annotations can be increased. Furthermore, gene loss events contribute to the variation in gene number among mammalian genomes. These gene losses have been reported to be involved with prominent physiological or metabolic adaptations (*Sharma et al., 2018*).

Clustering of orthologous genes using OrthoFinder suggests that more than 95% of genes were clustered into orthogroups and are associated among the eight *Myotis* species included in this analysis. By examining the *Myotis* species represented in each gene cluster, a total of 9,398 common orthologous gene clusters were identified for the eight genomes (Table S6). In our clustering analysis of New World *Myotis* species, distinct patterns emerged. Specifically, we identified 54, 65, and 101 orthologous groups unique to *M. vivesi*, *M. findleyi*, and *M. planiceps*, respectively (Fig. 4A). Similarly, a comparable proportion of exclusive orthogroups was observed in the cluster encompassing our focal species and the Old World *Myotis* species (Fig. 4B). These orthogroups probably represent evolutionarily young genes that have undergone divergence after gene duplication (*Assis & Bachtrog, 2015*: *Holland et al., 2017*). The paired association of orthogroups revealed a greater proportion of orthogroups shared between species that are more closely related.

The ML phylogenomic tree, constructed for 42 mammalian species, encompassing 40 bats (29 representatives from Yangochiroptera and 11 from Yinpterochiroptera), was based on the analysis of 247 sets of single-copy orthologous genes. This tree effectively elucidates the evolutionary history of Chiroptera. Our phylogenetic results align with previous research using a variety of transcriptomic and other data (*Teeling et al., 2005*; *Lei & Dong, 2016*; *Hawkins et al., 2019*). Yinpterochiroptera and Yangochiroptera received robust bootstrap support of 100, as determined from the concatenated alignment of orthogroups

(see Fig. 5). However, unlike earlier studies (*e.g.*, *Nikaido et al., 2020*), we do not observe the nesting of megabats within microbat lineages. Within the clade of microbats, which are included as part of the suborder Yangochiroptera, a distinct divergence is evident between the branches leading to the superfamily Vespertilionoidea and the branch leading to the family Phyllostomidae. The latter is the better-represented family, featuring a total of 13 members. Within the superfamily Vespertilionoidea, we successfully position the species that belong to the family Molossidae, with just two representative species, and the species grouped within the lineage of the family Vespertilionidae. Within the latter, our species of interest form a subclade with other *Myotis* members. However, the demarcation between the Old World and New World *Myotis* subclades is gradually becoming discernible. These results are consistent with findings from earlier investigations (*e.g.*, *Morales et al., 2019*). Nonetheless, the ongoing expansion of genomic resources for the *Myotis* genus and other bat species remains essential for gaining a comprehensive understanding of the evolution within this intriguing group.

## CONCLUSION AND PERSPECTIVES

Here we present the first genome assemblies of three endemic Mexican bat species from the genus *Myotis*. We performed the annotation of transposable elements as well as protein-coding genes. We assigned putative functions to these gene sets using various algorithms and databases. We predicted the presence of ncRNA elements within our genome sequences. Furthermore, we conducted orthologous gene clustering with other *Myotis* species, resulting in a substantial number of single-copy orthogroups, that can be utilized for future investigations involving phylogenetic reconstructions and gene family evolution. Integration of these genomic resources with developing and forthcoming datasets, as well as recently published studies for bats and other mammals, as the studies cited above should enable in-depth future studies aimed to detect species-specific genetic variants and molecular mechanisms that may potentially influence molecular adaptation underlying their evolution. We anticipate that the results of these multidisciplinary studies will have significant conservation applications, such as a better understanding of the genetic basis of system functions, including the immune system.

We believe these first genome assemblies of three endemic *Myotis* species from Mexico will be a useful resource to facilitate comparative genomic investigations in future studies, focusing on understanding how bats and other mammals have managed to survive and diversify in their environments through their evolutionary mechanisms, a previous bat studies have focused on acquiring and comparing complete genomes to establish associations between protein coding changes and specific adaptations (*Teeling et al., 2018*; *Jebb et al., 2020*) and to produce notable adaptation studies, including molecular adjustments to feeding habits (*Gutiérrez-Guerrero et al., 2020*), olfactory receptors (*Yohe et al., 2021*), echolocation (*Marcovitz et al., 2019*; *Wang et al., 2021*), and immune system (*Jebb et al., 2020*; *Moreno-Santillan et al., 2021*), among others. Future studies should analyze all the available Vespertilionidae genomes in a comparative manner to discover ecological and evolutionary patters to explain the adaptive radiation of this genus, and to understand why it is so successful.

## ACKNOWLEDGEMENTS

We extend our gratitude to Dr. Joaquin Arroyo-Cabrales for generously providing the valuable tissue samples of *M. planiceps*. Our sincere thanks are due to Dr. Anahí Martınez-Cárdenas for her indispensable guidance in utilizing bioinformatics programs. We express our gratitude to Dr. Diana D. Moreno-Santillán for granting us access to the gene annotations of various bat species. We acknowledge Ing. Rodrigo García for facilitating computational access and providing assistance. Special recognition goes to the individuals who lent their unwavering support during the field trips, including Dr. Veronica Zamora-Gutiérrez, Dr. M. Cristina Mac Swiney G., MSc Kenia Reyes Ochoa, MSc Elizabeth Castro-Salas, MSc Issachar L. López-Cuamatzi, and MSs Silvino E. Hernández-Cárdenas.

### Funding

This project was funded by CONAHCyT with the project: Ciencia de Frontera No. 15307. Edgar G. Gutiérrez received a scholarship from CONAHCyT through grant No. CVU: 891943. The funders had no role in study design, data collection and analysis, decision to publish, or preparation of the manuscript.

### Grant Disclosures

The following grant information was disclosed by the authors:
CONAHCyT with the project: Ciencia de Frontera No. 15307.
CONAHCyT: scholarship no. CVU: 891943.

### Competing Interests

Luis E. Eguiarte is an Academic Editor for PeerJ.

### Author Contributions

- Edgar G. Gutiérrez conceived and designed the experiments, performed the experiments, analyzed the data, prepared figures and/or tables, authored or reviewed drafts of the article, and approved the final draft.
- Jesus E. Maldonado conceived and designed the experiments, performed the experiments, prepared figures and/or tables, authored or reviewed drafts of the article, and approved the final draft.
- Gabriela Castellanos-Morales conceived and designed the experiments, performed the experiments, prepared figures and/or tables, authored or reviewed drafts of the article, and approved the final draft.
- Luis E. Eguiarte conceived and designed the experiments, performed the experiments, prepared figures and/or tables, authored or reviewed drafts of the article, and approved the final draft.
- Norberto Martínez-Méndez conceived and designed the experiments, performed the experiments, prepared figures and/or tables, authored or reviewed drafts of the article, and approved the final draft.

- Jorge Ortega conceived and designed the experiments, performed the experiments, prepared figures and/or tables, authored or reviewed drafts of the article, and approved the final draft.

## Animal Ethics

The following information was supplied relating to ethical approvals (i.e., approving body and any reference numbers):

Secretaria del Medio Ambiente y Recursos Naturales (SEMARNAT) License. No. SGPA/DGVS/07992/20.

## Field Study Permissions

The following information was supplied relating to field study approvals (i.e., approving body and any reference numbers):

Secretaria del Medio Ambiente y Recursos Naturales (SEMARNAT), Mexico.

## Data Availability

The genome assembly sequences are available at NCBI Bioproject number: PRJNA1021227; for *M. vivesi* (JAWPEG000000000 and BioSample SAMN37984258); *M. planiceps* (JAYKON000000000 and BioSample SAMN37987054); and *M. findleyi* (JAWQEE000000000 and BioSample SAMN37985987).

## Supplemental Information

Supplemental information for this article can be found online at http://dx.doi.org/10.7717/peerj.17651#supplemental-information.

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
