# Peer review of "Unraveling genomic features and phylogenomics through the analysis of three Mexican endemic Myotis genomes"

_PeerJ, doi:10.7717/peerj.17651_

## Round 0.1 · original submission · Major Revisions

Dear authors,

We extend our sincere appreciation for your meticulous review of the manuscript entitled "Unraveling genomic features and phylogenomics through the analysis of three Mexican endemic Myotis genomes." Based on the reviewers' insightful comments, we have deliberated on the manuscript and concluded that major corrections are necessary before it can be accepted for publication.

Key points highlighted by the reviewers that require immediate attention include:

- Deposition of Raw Data in a public database.

- Reference Genome Quality and Assembly Details: providing additional insights into the quality of the Myotis yumanensis reference genome used for anchoring the genome assemblies of the three endemic Myotis species.

- Additional Analyses: conducting a simple genome assembly with the Illumina data without the reference genome for scaffolding.


We eagerly anticipate receiving your revisions and express our sincere gratitude for selecting PeerJ as the platform to present your significant research.

Warm regards,

Armando Sunny

Reviewer 1 ·

Basic reporting

This manuscript by Gutierrez et al. has the potential to be a valuable addition to the literature related to chiropteran genomics. Myotis is the most species rich genus of bats, exhibiting multiple recent radiations and has the potential to be a gold mine for researchers interested in mammalian speciation dynamics. Thus, I was excited to see this manuscript and happy to review it.

While the manuscript is generally well-written, all that excitment is tempered a bit but some serious questions about the data analysis and interpretation. There are also multiple textual errors throughout the text and in the figures. All of those concerns are detailed below. In the end, I recommend a substantial revision that includes the addition of some additional analyses using the M. myotis genome assembly.

Line 84 - Change constrained to contraint.

L85 - Change to orders

L93 - Consider changing this to 'species rich' due to the multiple definitions of diversity. Diversity could refer to the large number of species but also to phenotypic diversity, and other groups (e.g. Phyllostomids) are much more diverse if you're using that definition.

L182 - "a coverage of 30X." I don't understand this statement. A read isn't characterized by it's coverage. Coverage is how often a given position in the genome is covered by the reads. If a position has 30X coverage, 30 reads provide information on that position. A read is a single bit of information on that position and doesn't have a coverage statistic. Do you mean the number of copies of that particular read?

L194 - Change scaffold to scaffolder

L194 and onward - Not familiar with this program but my reading indicates that the assemblies described in this paper are not de novo assemblies. By using the M. yumanensis assembly as a starting point, this sounds more like a reference-based assembly. You never actually say that these are de novo assemblies but I assumed they were when I started reading and it would be good to indicate this much earlier in the manuscript. In fact, the ntJoin paper refers to the ntjoin output as reference-based. Reference-based assemblies are valuable but, because they're based on existing assemblies, there is the chance that some regions not present in the reference but present in the non-reference species may be missed. And vice versa. I could see how this method may also miss lineage-specific rearrangements. How were these possibilities evaluated? Was the 'no_cut=True' option that minimizes these possibilities invoked? https://doi.org/10.1093/bioinformatics/btaa253

L196-197 - Why use yumanensis when lucifugus is available? Is it a better assembly? I also understand wanting to use a New World species. However, the M. myotis assembly is the best Myotine assembly available and could be a valuable tool. Was it tried?

L303 - change to '...and have a Phred quality...'

L311 and so on - See my concerns about reference assemblies described above. It's possible that these are all similar in size because they're all from relatively closely related species but it could be because they're all based on M. yumanensis. What's the size of the yumanensis assembly? M planiceps is 300 Mb larger than the other two. The good news is that this suggests that some de novo regions were identified but was this confirmed? Where are the extra 300 Mb coming from? The numbers for the M. planiceps assembly are very bad compared to the other two. That needs to be discussed somewhere.

L320 - Wow. This is exceptionally low compared to the other two. Why? This is making me think that something went very wrong with the M. planiceps assembly.

L330-331 - This is odd. This assembly was 300 Mb larger than the other two. Given the direct relationship between genome size and TE accumulation, one would expect there to be more TE content in this species. Instead, the opposite is observed. That needs some sort of explanation. It's clear that the planiceps assembly isn't as good as the other two. I suspect the extra 300Mb is derived from misassemblies. Why this assembly and not the others? Was an attempt made to identify misassemblies?

L380 - These three non-endemics are odd choices. Why not use the highest quality Myotis assembly available, M. myotis? Instead, the authors chose three assembies of lower quality. It couldn't be because of the relatively large phylogenetic distance, as both M. myotis and M. davidii are both Old World species and therefore equally distantly related to the New World taxa under examination (assuming our understanding of the Myotis invasion of North America is correct and according to your own tree in Figure 5). This strange choice needs to be explained or the high-quality M. myotis assembly should be included.

L404 - Can you really call these draft genomes? Technically they are, according to the definition of 'draft genome'. You should really include reference-based in the description, though.

L412 - Change Vespertilionidae to the adjective 'vespertilionid' or to 'vesper bat'.

L465 - Change to Platt.

L469 and so on - I'm not sure this paragraph is necessary. The identification of Helitrons is completely expected and not surprising, given the position of these three species in the family Vespertilionidae. I recommend that it be removed.

L471 - Capitalize Helitrons.

L481-482 - "of which the largest number of sequences were in the three species of the
482 snoRNA type,". This is a badly written sentence. I'm not sure how to recommend fixing it without potentially changing the meaning.

L575 - Change ration to radiation.

L575 - I don't understand what is being communicated by, "the more speciose in mammals".

Figure 2 - Change Repeated landscape to Repaat landscapes.

Figure 2 - The y-axis is mislabeled. It looks like these are actual base counts and not genomic proportions.

Figure 5 - Mus musculus and R. aegyptiacus are both misspelled in the tree.

Experimental design

See content above.

Validity of the findings

See content in the previous boxes.

Annotated reviews are not available for download in order to protect the identity of reviewers who chose to remain anonymous.

Reviewer 2 ·

Basic reporting

No comments

Experimental design

The manuscript describes the sequencing, assembly, and annotation of genomes of three endemic Myotis bats from Mexico. While the methods and results are detailed, I could not find the details of the deposition of the raw data in a public database. If this is not done already then I request the authors to make the data available and provide the accession numbers in the manuscript.

Validity of the findings

Please provide details about the Myotis yumanensis reference genome quality that was used to anchor the genome assemblies of the three endemic Myotis species.

I have some reservations about the N50 statistic. The N50 of the three genomes is anywhere between 86 to 93 Mb. However, the number of scaffolds varies between ~56,000 to 250,000. This is very ambiguous. The N50 would suggest a very high-quality genome assembly and there should be only a few hundred scaffolds in this case. The size of the longest scaffold is around 200 Mb suggestive of a chromosomal-level assembly.

The authors should provide details about the number of scaffolds that are less than 1000 bp.


I recommend that the authors do a simple genome assembly with the Illumina data without the reference genome for scaffolding and test the quality of the assembly.

---

## Round 0.2 · Minor Revisions

Dear authors,

The corrections you made were very complete, as commented by reviewer 1, improving the manuscript considerably. For this reason, only final remaining minor corrections are needed for the manuscript to be accepted.

Best regards,

Armando Sunny

Reviewer 1 ·

Basic reporting

The authors did an admirable job addressing my earlier concerns. I have only a few minor suggestions.

L191 - 193. Change to "Reads were filtered to a miniumum length of 150 bp and sequence coverage for all species was >30X.
L363 - remove the extra 's' in 'assembliess'
L486 - a space is needed between bat and have
The legend for Figure 2 is still incorrect. It should be Repeat landscape, not Repeated landscape.

Experimental design

See above.

Validity of the findings

See above.

---

## Round 0.3 · accepted · Accept

Dear Authors,

Thank you for your thorough and meticulous revisions in response to the reviewers' comments. After careful consideration, we are pleased to inform you that your manuscript is now accepted for publication.

We greatly appreciate your choice of PeerJ for sharing your important and interesting work.

Best regards,

Armando Sunny